# Associations between 10-second standing balance tests and mild cognitive impairment in older adults in Colombia: A cross-sectional national survey

Patricia García-Garro[1], Olga L. Sarmiento[2], Gary O'Donovan[2]*

**1** Facultad de Educación y Pedagogía, Universidad del Valle, Cali, Colombia, **2** Facultad de Medicina, Universidad de los Andes, Bogotá, Colombia

* g.odonovan@uniandes.edu.co

## Abstract

Very little is known about the associations between balance and cognition in Latin America. Therefore, the objective of this study was to investigate associations between 10-second standing balance tests and mild cognitive impairment using data from a large nationally representative study of older adults in Colombia. The sample included 23,443 community-dwelling adults aged 71 (8) years [mean (SD)] from the National Survey of Health, Wellbeing and Ageing in Colombia. Trained interviewers collected the data in 2015. Mild cognitive impairment was defined as a score of 12 or less out of 19 on the shorter version of the mini-mental state examination used and validated in the region. Balance was assessed using three increasingly difficult standing balance tests: the feet-together test, the semi-tandem test, and the tandem test. Logistic regression models were adjusted for age, sex, education, smoking, civil status, diabetes, and physical activity. The prevalence of mild cognitive impairment was 20% in 23,443 participants who attempted the feet-together test, 16% in 20,943 who attempted the semi-tandem test, and 15% in 19,527 who attempted the tandem test. The odds ratio (95% confidence interval) for mild cognitive impairment was 0.43 (0.39, 0.48) for a successful feet-together test, 0.53 (0.46, 0.60) for a successful semi-tandem test, and 0.63 (0.57, 0.70) for a successful tandem test after adjusting for potential confounders [the p-value for each model was highly significant (all p < 0.001)]. There is a need for simple screening tools in Colombia and other countries in Latin America with limited resources. This study suggests that even the simplest of balance tests could be used to identify older adults at risk of mild cognitive impairment in Colombia. Longitudinal studies are needed to confirm these novel and important findings.

**Data availability statement:** The data used in this analysis are available upon reasonable request, depending on a user agreement. Please email the Colombian Ministry of Health to request access to the data: repositorio@min-salud.gov.co. The original name of the study is in Spanish: Salud, Bienestar & Envejecimiento (SABE Colombia). The analysis plan is available from the corresponding author.

**Funding:** The authors received no specific funding for this work.

**Competing interests:** The authors have declared that no competing interests exist.

## Introduction

Cognitive impairment and dementia are estimated to affect approximately 57 million people around the world and the number of cases is expected to increase to more than 150 million by 2050 [1]. Data from the United Nations suggest that the rising burden of non-communicable diseases in Latin America is due to population growth and population ageing [2]. Furthermore, data from the Global Burden of Disease Study suggest that the projected trends in population growth and population ageing are such that the proportion of people living with dementia is expected to increase by around 75% by 2050 in western Europe and by more than 250% in Latin America [1]. In Colombia alone, it is currently estimated that the cost of dementia is around US$ 1,158 million per year [3]. The human and economic costs of dementia are such that there is great interest in identifying simple tests of physical and cognitive function, including physical performance tests [4–6]. Cross-sectional studies suggest that simple balance tests can be used to predict cognitive decline in older adults in the United States [7]. Longitudinal studies also suggest that balance is associated with dementia in older adults in the United States and the United Kingdom [8,9]. However, very little is known about the associations between simple physical performance tests and cognition in Latin America [10]. It is plausible that balance is associated with cognition [11–13]. For example, good balance depends on feedback from the vestibular system and studies have shown that vestibular dysfunction is associated with worse balance and cognitive decline [13].

It is important to identify simple screening tools in countries in Latin America with limited resources because many commonly used screening tools are costly and difficult to implement [14]. It is particularly important to gather evidence from Colombia because the country is so diverse and so difficult to understand after suffering more than 50 years of civil war [15,16]. The National Survey of Health, Wellbeing and Ageing in Colombia is a nationwide study of community-dwelling adults aged 60 years or older (SABE Colombia, in Spanish) [17–19]. The objective of the present analysis was to investigate associations between 10-second standing balance tests and mild cognitive impairment in participants in SABE Colombia. It would have important implications for policy and practice in Colombia if a simple screening tool were found to predict cognitive decline because a five-year delay in onset may dramatically reduce the prevalence of dementia [20].

## Materials and methods

### Ethics statement

Ethical approval was obtained from institutional review boards of *Universidad de Caldas* (CBCS-021–14) and *Universidad del Valle* (09–014 and 011–015) and all participants gave written informed consent.

### Participants

The present analysis was conducted in agreement with the STROBE Statement and contains items that should be included in reports of cross-sectional studies [21].

The National Survey of Health, Wellbeing and Ageing in Colombia is a cross-sectional study that is described in detail elsewhere [17,18]. Briefly, the target population was all adults aged 60 years or older living in households. Participants were selected using a multistage area probability sampling design and there were four selection stages: municipalities, blocks, housing units, and households [22]. People living in all regions of Colombia were invited to participate, including people living in the four large cities of Bogotá, Medellín, Cali, and Barranquilla [22]. The response rate was around 62% in urban areas, around 77% in rural areas, and around 70% overall [17]. Data were collected across all departments (that is, states) and the final sample of 23,694 men and women was deemed to be representative of the population of older adults living in households in Colombia [17]. Trained interviewers conducted face-to-face interviews in the participant's home between April and September 2015 [17].

### Dependent variable

The dependent variable was mild cognitive impairment. The versions of the mini-mental state examination (MMSE) used in Latin America and the Caribbean are shorter than the original version in an attempt to reduce the low literacy bias [23]. The shorter version of the MMSE used in SABE Colombia has six questions and participants were asked: to state the date and the day of week (4 points); to repeat and remember three words (3 points); to state in reverse order the numbers 1, 3, 5, 7, 9 (5 points); to take a piece of paper in their right hand, fold it in half using both hands, and put it on their lap (3 points); to reiterate the three words given earlier (3 points); and, to copy a drawing of two overlapping circles (1 point). A score of 12 or less out of 19 was deemed to be indicative of mild cognitive impairment in SABE Colombia [17]. The shorter version of the MMSE used in SABE Colombia has been validated in a study of 1,301 adults aged 60 years or older living in households in Chile [24]. The prevalence of mild cognitive impairment was 10.7% using the threshold of 12 or less out of 19 in the shorter version of the MMSE and 8.1% using the threshold of 6 or more out of 33 in the criterion measure [24], which was the Short Portable Mental Status Questionnaire [25].

### Independent variable

The independent variable was 10-second balance test performance. The trained interviewer informed the participant that they would like them to attempt three increasingly difficult balance tests. The interviewer encouraged the participant to attempt each test but emphasised that the participant did not have to attempt a test if they thought it might be unsafe to do so. Fig 1 shows the position of the feet during each of the three balance tests. Before the feet-together test, the interviewer said: "I would like you to try to stand with your feet together for 10 seconds; please watch while I do it. You can use your arms, bend your knees, or move your body to maintain your balance. Try to maintain the position until I tell you to stop." Before the semi-tandem test, the interviewer said: "Now I would like you to try to stand with the heel of one foot touching the ball of the other foot and to maintain the position for 10 seconds; please watch how I do it." Before the tandem test, the interviewer said: "Now I would like you to place one foot in front of the other, with the heel of one foot touching the tip of the toes of the other foot for 10 seconds; please watch how I do it." A test was regarded as successful if the participant maintained the required position for 10 seconds.

### Covariates

The analyses were adjusted for a range of covariates that may be associated with cognitive decline in older adults, including age, sex, education, smoking, civil status, diabetes, and physical activity [11]. The trained interviewers asked participants about the highest level of education they had achieved, and we created three groups: no education; some primary education; and, some secondary education or more. Participants were also asked about cigarette smoking, and we created three groups: never, former smoker, and current smoker. Participants were asked about their current civil status, and we created two groups: not married or with partner; and, married or with partner. Participants were also asked whether a doctor or a nurse had ever told them they had diabetes. Finally, the interviewers asked the participants about sport and

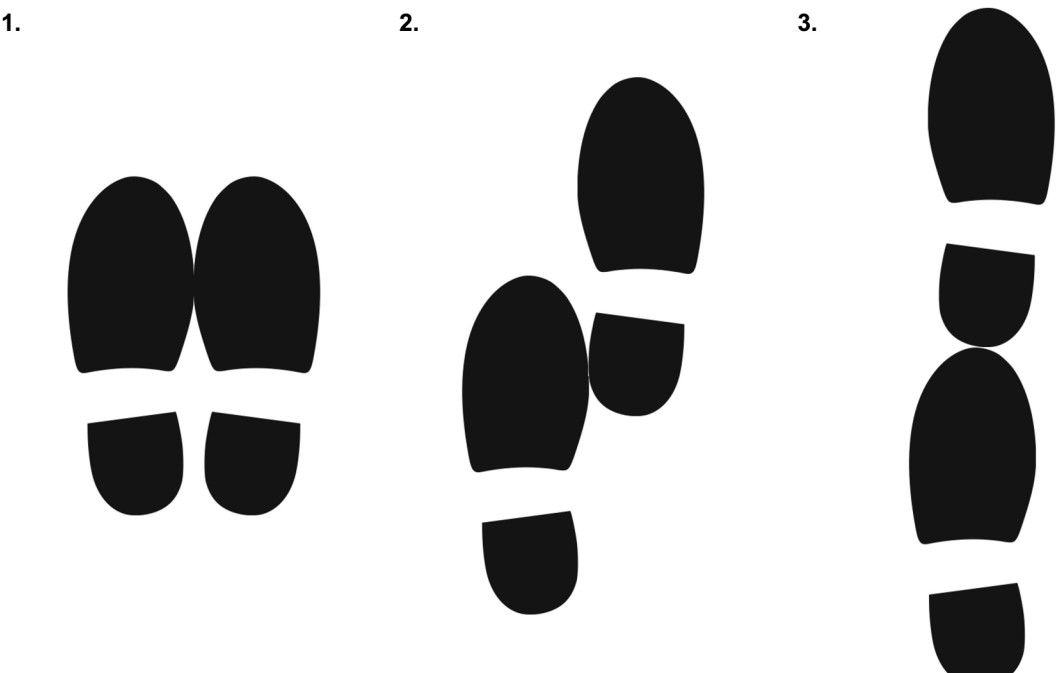

**Fig 1. The position of the feet during (1) the feet-together balance test, (2) the semi-tandem balance test, and (3)] the tandem balance test.**

exercise and about walking. Participants were regarded as physically active if they reported taking part in vigorous sport or exercise at least three times a week or walking between 9 and 20 blocks at least three times per week.

## Statistical analyses

Logistic regression was used to investigate associations between balance and mild cognitive impairment. The histogram of the residuals resembled the normal distribution but was slightly skewed towards higher MMSE scores. All logistic regression models were adjusted for age, sex, education, smoking, civil status, diabetes, and physical activity. Age was modelled as a continuous variable. All other covariates were modelled as categorical variables. The likelihood-ratio chi-squared test for each model was highly significant (all $p < 0.001$). Odds ratios and 95% confidence intervals were calculated for each model. We used the Wald statistic to test the robustness of each model. We also fitted logit models to check for collinearity among the covariates. All analyses were performed using Stata MP version 18.0 for Mac (StataCorp, Texas, USA).

## Results

Table 1 shows participants' characteristics according to balance test. The present analysis included data from 23,443 of 23,694 (99%) participants in SABE Colombia. All participants attempted the feet-together test, 20,943 participants attempted the semi-tandem test, and 19,527 participants attempted the tandem test and had complete data on the covariates. The prevalence of mild cognitive impairment was 20% in those who attempted the feet-together test and was lower in those who attempted the semi-tandem test and those who attempted the tandem test. Age, sex, and smoking were similar across the groups. The proportion with no education decreased slightly across the groups. The proportion married or with partner increased slightly across the groups. Finally, the proportion with diabetes increased while the proportion who were physically active decreased.

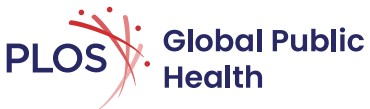

**Table 1. Participants' characteristics according to 10-second balance test.**

| | Feet-together test (n = 23,443) | Semi-tandem test (n = 20,943) | Tandem test (n = 19,527) |
|---|---|---|---|
| Successful balance test | | | |
| No, n (%) | 2,532 (10.80) | 1,428 (6.82) | 3,662 (18.75) |
| Yes, n (%) | 20,911 (89.20) | 19,515 (93.18) | 15,865 (81.25) |
| MMSE score, mean (SD) | 14.96 (3.89) | 15.35 (3.48) | 15.53 (3.32) |
| Mild cognitive impairment | | | |
| No, n (%) | 18,853 (80.42) | 17,528 (83.69) | 16,670 (85.37) |
| Yes, n (%) | 4,590 (19.58) | 3,415 (16.31) | 2,857 (14.63) |
| Age in years, mean (SD) | 70.79 (8.19) | 70.02 (7.62) | 69.55 (7.33) |
| Sex | | | |
| Male, n (%) | 10,008 (42.69) | 9,138 (43.63) | 8,665 (44.37) |
| Female, n (%) | 13,435 (57.31) | 11,805 (56.37) | 10,862 (55.63) |
| Education | | | |
| None, n (%) | 5,192 (22.15) | 4,482 (21.40) | 4,036 (20.67) |
| Some primary, n (%) | 13,373 (57.04) | 11,993 (57.26) | 11,218 (57.45) |
| Some secondary or more, n (%) | 4,878 (20.81) | 4,468 (21.33) | 4,273 (21.88) |
| Cigarette smoking | | | |
| Never smoked, n (%) | 11,266 (48.06) | 9,960 (47.56) | 9,260 (47.42) |
| Former smoker, n (%) | 9,684 (41.31) | 8,665 (41.37) | 8,066 (41.31) |
| Current smoker, n (%) | 2,493 (10.63) | 2,318 (11.07) | 2,201 (11.27) |
| Civil status | | | |
| Not married or with partner, n (%) | 10,988 (46.87) | 9,474 (45.24) | 8,628 (44.18) |
| Married or with partner, n (%) | 12,455 (53.13) | 11,469 (54.76) | 10,899 (55.82) |
| Diabetes | | | |
| No, n (%) | 19,583 (83.53) | 17,611 (84.09) | 16,495 (84.47) |
| Yes, n (%) | 3,860 (16.47) | 3,332 (15.91) | 3,032 (15.53) |
| Physically active | | | |
| No, n (%) | 11,182 (47.70) | 9,140 (43.64) | 8,138 (41.68) |
| Yes, n (%) | 12,261 (52.30) | 11,803 (56.36) | 11,389 (58.32) |

Data are from the National Survey of Health, Wellbeing and Ageing in Colombia. Participants were community-dwelling adults aged 60 years or older. MMSE refers to the shorter version of the mini-mental state examination used in the survey. Mild cognitive impairment was defined as a score of 12 or less out of 19 on the MMSE.

Table 2 shows odds ratios for mild cognitive impairment according to successful 10-second balance test performance. The odds ratio (95% confidence interval) for mild cognitive impairment was 0.43 (0.39, 0.48) for the feet-together test, 0.53 (0.46, 0.60) for the semi-tandem test, and 0.63 (0.57, 0.70) for the tandem test after adjusting for age, sex, education, smoking, civil status, diabetes, and physical activity. The p-value for each model was highly significant (all $p < 0.001$). The Wald statistic for each model was also highly significant (all $p < 0.001$), which suggests that the models are robust and that sound inferences can be made. There was no evidence of collinearity. A total of 4,590 cases of mild cognitive impairment were identified using the feet-together test and no additional cases were identified using the other tests.

## Discussion

The objective of this study was to investigate associations between 10-second balance tests and mild cognitive impairment in older adults in Colombia. We used data from a nationally representative study of older adults and we found that

**Table 2. Odds ratios for mild cognitive impairment according to successful 10-second balance test performance.**

| Balance test | N/cases (% cases) | Odds ratio (95% confidence interval) |
|---|---|---|
| Feet-together test* | 23,443/4,590 (20%) | 0.43 (0.39, 0.48)† |
| Semi-tandem test* | 20,943/3,415 (16%) | 0.53 (0.46, 0.60)† |
| Tandem test* | 19,527/2,857 (15%) | 0.63 (0.57, 0.70)† |

Mild cognitive impairment was defined as a score of 12 or less out of 19 on the shorter version of the mini-mental state examination used in SABE Colombia. All analyses were adjusted for age, sex, education, smoking, civil status, diabetes, and physical activity.

*The Wald statistic for each model was highly significant (all p < 0.001).

†The p-value for each model was also highly significant (all p < 0.001). There was no evidence of collinearity among the covariates.

the risk of mild cognitive impairment was around 60% lower in those who successfully completed the feet-together test, around 50% lower in those who successfully completed the semi-tandem test, and around 40% lower in those who successfully completed the tandem test after adjusting for a range of potential confounders. All cases of mild cognitive impairment were identified using the first test, the feet-together test. This cross-sectional study suggests that even the simplest of balance tests can be used to identify older adults at risk of mild cognitive impairment in Colombia. Longitudinal studies are needed to confirm these novel and important findings.

The associations of balance with mild cognitive impairment and dementia have been assessed in a small number of studies of older adults [7–10], mainly in Europe and North America [7–9]. For example, Meunier and colleagues investigated associations between standing balance tests and cognition in older adults in the United States [7]. In a cross-sectional analysis of 3,184 men and women, Meunier and colleagues found that the 10-second feet-together test, the 10-second semi-tandem test, and the 10-second tandem test predicted cognitive function, as assessed using a modified mini-mental state examination and a digit symbol substitution test [7]. In a longitudinal analysis of 4,811 men and women followed for six years, they found that the five-second tandem test predicted cognitive impairment after adjusting for confounders [7]. Stephan and colleagues investigated associations between the 10-second semi-tandem test and the risk of dementia in older adults in the United States and the United Kingdom who were followed for up to 15 years [9]. The hazard ratio (95% confidence interval) for dementia was 1.58 (1.26, 1.96) in 5,658 men and women in the United States and 1.97 (1.24, 3.14) in 3,667 men and women in the United Kingdom after adjustment for demographic factors [9]. To the best of our knowledge, the present study is the first study of the associations between balance and cognition in Colombia and the largest study of its kind in Latin America. The results of the present study are biologically plausible. For example, vestibular dysfunction would explain why hearing loss may be a risk factor for dementia in middle-aged adults [11] and why poor balance may be a risk factor for dementia in older adults [7,9]. The 'hearing system' of the outer ear and the middle ear and the 'balance system' of the inner ear are both connected to the brain via the vestibulocochlear nerve. Good balance depends upon signals and feedback from the vestibular system and the brain, and studies have shown that vestibular function declines with age [12] and that vestibular dysfunction is associated with worse balance and cognitive decline [13].

The present study may have important implications for policy and practice in Colombia and other countries in Latin America with limited resources. Early detection of cognitive impairment can reduce the burden of dementia by permitting timely action from health care providers, family members, and patients [26]. A recent evaluation of the current state of dementia prevention and care in Colombia made several recommendations for policy and practice in the country, including: the development of a national dementia plan in keeping with World Health Organization guidelines; greater emphasis on education and other social determinants of health; implementation of education campaigns to reduce the



stigma associated with cognitive decline; greater access to health care; a multisectoral approach to primary and secondary prevention; and the promotion of research into dementia in the country [27]. The present study is important because it suggests that cheap and convenient balance tests could be used to identify those older adults in Colombia who would benefit from the policies and practices mentioned in the national report on dementia prevention and care.

This study has strengths and limitations. The National Survey of Health, Wellbeing and Ageing in Colombia is the largest study of its kind in Colombia [19]. The present analysis included data from 99% of participants in SABE Colombia and the models were adjusted for a range of potential confounders. The cross-sectional design is the main limitation. Temporality is the major concern when inferring causality [28] and cohort studies with long follow-up times are required to confirm the associations between balance tests and mild cognitive impairment observed in the present study [29]. The data were collected in 2015 and may not be representative of the current population of older adults in Colombia. Static balance tests were used in the present study and more research is required to determine whether dynamic balance tests predict mild cognitive impairment and dementia in older adults in Colombia. The screening tool used in the present study is valid [30], but the MMSE does not provide a clinical diagnosis of cognitive impairment. Finally, self-reported variables may be subject to biases.

## Conclusions

There is a need for simple screening tools in Colombia and other countries in Latin America with limited resources. Very little was known about the associations between balance and cognition in Colombia. We used data from a nationally representative study of older adults and we found that the risk of mild cognitive impairment was lower in those who completed the feet-together test, the semi-tandem test, or the tandem test. This cross-sectional study suggests that even the simplest of balance tests can be used to identify older adults at risk of mild cognitive impairment in Colombia. Longitudinal studies are needed to confirm these novel and important findings.

## Author contributions

**Conceptualization:** Gary O'Donovan.

**Data curation:** Olga L. Sarmiento, Gary O'Donovan.

**Formal analysis:** Gary O'Donovan.

**Methodology:** Gary O'Donovan.

**Project administration:** Gary O'Donovan.

**Supervision:** Olga L. Sarmiento, Gary O'Donovan.

**Writing – original draft:** Patricia García-Garro, Olga L. Sarmiento, Gary O'Donovan.

**Writing – review & editing:** Patricia García-Garro, Olga L. Sarmiento, Gary O'Donovan.

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
