## [Decision Letter · Decision Letter 0]

6 Jun 2025

PGPH-D-25-00809

Associations between 10-second balance tests and mild cognitive impairment in older adults in Colombia: a national survey

Dear Dr. O'Donovan,

Thank you for submitting your manuscript to PLOS Global Public Health. After careful consideration, we feel that it has merit but does not fully meet PLOS Global Public Health’s publication criteria as it currently stands. Therefore, we invite you to submit a revised version of the manuscript that addresses the points raised during the review process.

We look forward to receiving your revised manuscript.

Kind regards,

Professor Razak Gyasi, PhD, PD

Academic Editor

Journal Requirements:

1. We noticed you have some minor occurrence of overlapping text with the following previous publication(s), which needs to be addressed:

- https://doi.org/10.1186/s11556-022-00307-y

-doi: 10.1038/s41598-020-74822-2

In your revision ensure you cite all your sources (including your own works), and quote or rephrase any duplicated text outside the methods section. Further consideration is dependent on these concerns being addressed.

Additional Editor Comments (if provided):

Reviewers' comments:

Reviewer's Responses to Questions

**Comments to the Author**

1. Does this manuscript meet PLOS Global Public Health’s publication criteria? Is the manuscript technically sound, and do the data support the conclusions? The manuscript must describe methodologically and ethically rigorous research with conclusions that are appropriately drawn based on the data presented.

Reviewer #1: Yes

Reviewer #2: Yes

2. Has the statistical analysis been performed appropriately and rigorously?

Reviewer #1: Yes

Reviewer #2: Yes

3. Have the authors made all data underlying the findings in their manuscript fully available (please refer to the Data Availability Statement at the start of the manuscript PDF file)?

Reviewer #1: No

Reviewer #2: Yes

4. Is the manuscript presented in an intelligible fashion and written in standard English?

Reviewer #1: Yes

Reviewer #2: Yes

5. Review Comments to the Author

Reviewer #1: Data availability statement as currently written, does not fully meet PLOS Global Public Health’s open data policy. Thus, compliance statement could have been written as 'The data underlying this study are owned by the Colombian Ministry of Health and cannot be made publicly available due to ethical and legal restrictions protecting participant privacy. Data are available on reasonable request from the Colombian Ministry of Health (repositorio@minsalud.gov.co) for qualified researchers who meet the criteria for access to confidential data.'

For more comments on this and other sections, see separately attached document.

Reviewer #2: Manuscript Number: PGPH-D-25-00809 Review Report

Title: Associations between 10-second balance tests and mild cognitive impairment in older adults in Colombia: a national survey

Title

Strength

• It clearly indicated; if there were an association between physical balance and cognitive health in older adults

• Study area and study participants clearly indicated; older adults in Colombia.

• Type of the study is clearly indicated; a national survey that is cross sectional design

Weakness

• 10-second balance tests are not clear; what kind of test is conducted? Specify the exact test or describe it more clearly

• A national survey” is vague—better if it is described as, Cross-sectional analysis or Findings from a national health survey.

• Study design is not clearly indicated “A cross-sectional national survey.

Overall Recommendation: suggested title

Association between single-leg 10-second balance test performance and mild cognitive impairment among older adults in Colombia: A cross-sectional national survey

Abstract

Strength

• It is well-structured; Objectives, Methods, Results, Conclusions clearly written

• It addresses the under researched area particularly in Latin America including Columbia. The link between physical and cognitive function.

• The objective is clearly stated and addresses a gap in knowledge (lack of data in Latin America)

• Study participants; clearly indicated that is older

• The result is reported clearly reported; confidence intervals are appropriate

• The conclusion is concise and highlights the practical implications.

Weakness

• What is the importance of studying this? Why balance test is undertaken? better if one sentence introduction is incorporated

• What was time of the survey? study period is not indicated

• Is the standard aligning with you counties? Can you elaborate or indicate any national guideline or reference in context of Columbia to classify Mild cognitive impairment?

• How the study reported? Cut off point for reporting the finding not indicated under method section (95% CI; P Value)

• How was the test conducted? Separately or at an interval? One person undertakes the three test simultaneously or only one test?

• The result reported only the prevalence of one test; that is feet-together test. What was the overall prevalence of the total sample? Better to include the outcome of the three tests from the total.

• What was the meaning of success in terms of balance test? completing the test without support for full 10 seconds

• Under conclusion which test is more recommended for further study? What is the prevention or intervention needed to be undertaken?

Overall Recommendation: Abstract

• Minor revisions suggested.

Introduction

Strength

• The introduction is well structured; it indicted global and nation burden of dementia. It give emphasize on the importance of simple predictive tools and finally identifies a gap in research specific to Latin America.

Weakness/Area of improvement

Why it is common in Latin America Specially Columbia? It is disproportionally affected than others; are there additional factors that influence it? Support it by additional literatures.

• What prior evidence suggests a link between physical performance, balance, and cognition, to strengthen rationale? Why this specific test could enhance the justification.

Materials and Methods

It provides sufficient information on the sampling, data collection, outcome variable, and ethical procedures

Sampling design is properly described; multistage probability sampling

Ethical approval was obtained from appropriate body and shows compliance with ethical standards.

Weakness/Areas of improvement

• The data were old; why it is not published so far? Even if it is recent national survey and may have relevance at this time.

• It does not show the total number of participants included in the analysis in this section.

Overall Recommendation: Materials and Methods

• Minor refinements for flow and completeness; especially inclusion of sample size and clarity around classification of MCI

Dependent variable and Independent variable

• dependent variable is well-explained

• The MMSE tool is validated by Chilean study; which increase credibility of the study

• The independent variable is described step by step, including instructions given to participants.

• What lower score <12 or High score 19 indicate? Poor or better cognitive performance? better if clearly explained

• How the number of test reported? Passed/fail; how they were classified passed 0, 1, 2, or all 3 balance tests?

•

Statistical Analyses

• It is well-structured and methodologically sounds. It demonstrates an appropriate analytical approach and thoughtful consideration of confounders.

Results

• It is clearly written and effectively summarizes the findings.

• Why fewer participants attempted the semi-tandem and tandem tests. What is the reason behind it? for example only 19,527 participants attempted the tandem test

• Odds ratios with 95% confidence intervals properly reported

Area of improvement

• It should be clear the reason why fewer participants attempted the semi-tandem and tandem tests.

• clarifications of test attempts vs. successful completion should be given to make the finding coherent

Discussion

• It is written in comprehensive manner, well-structured and relevance for policy.

• It contextualizes the findings within existing literature, addresses implications, and thoughtfully discusses strengths and limitations.

6. PLOS authors have the option to publish the peer review history of their article (what does this mean?). If published, this will include your full peer review and any attached files.

**Do you want your identity to be public for this peer review?** For information about this choice, including consent withdrawal, please see our Privacy Policy.

Reviewer #1: No

Reviewer #2: No

---

## [Decision Letter · Decision Letter 1]

8 Aug 2025

PGPH-D-25-00809R1

Associations between 10-second standing balance tests and mild cognitive impairment in older adults in Colombia: a cross-sectional national survey

Dear Dr. O'Donovan,

Thank you for submitting your manuscript to PLOS Global Public Health. After careful consideration, we feel that it has merit but does not fully meet PLOS Global Public Health’s publication criteria as it currently stands. Therefore, we invite you to submit a revised version of the manuscript that addresses the points raised during the review process.

We look forward to receiving your revised manuscript.

Kind regards,

Helen Howard

Staff Editor

Journal Requirements:

1. We noticed you have some minor occurrence of overlapping text with the following previous publication(s), which needs to be addressed:

- https://doi.org/10.1186/s11556-022-00307-y

-doi: 10.1038/s41598-020-74822-2

In your revision ensure you cite all your sources (including your own works), and quote or rephrase any duplicated text outside the methods section. Further consideration is dependent on these concerns being addressed.

Additional Editor Comments (if provided):

Reviewers' comments:

Reviewer's Responses to Questions

**Comments to the Author**

1. If the authors have adequately addressed your comments raised in a previous round of review and you feel that this manuscript is now acceptable for publication, you may indicate that here to bypass the “Comments to the Author” section, enter your conflict of interest statement in the “Confidential to Editor” section, and submit your "Accept" recommendation.

Reviewer #1: All comments have been addressed

Reviewer #2: All comments have been addressed

2. Does this manuscript meet PLOS Global Public Health’s publication criteria? Is the manuscript technically sound, and do the data support the conclusions? The manuscript must describe methodologically and ethically rigorous research with conclusions that are appropriately drawn based on the data presented.

Reviewer #1: Yes

Reviewer #2: Yes

3. Has the statistical analysis been performed appropriately and rigorously?

Reviewer #1: Yes

Reviewer #2: Yes

4. Have the authors made all data underlying the findings in their manuscript fully available (please refer to the Data Availability Statement at the start of the manuscript PDF file)?

Reviewer #1: Yes

Reviewer #2: No

5. Is the manuscript presented in an intelligible fashion and written in standard English?

Reviewer #1: Yes

Reviewer #2: Yes

6. Review Comments to the Author

Reviewer #1: The authors have adequately addressed the previously suggested comments and revisions. Therefore, I consider the manuscript acceptable for publication once the remaining minor redundancies are refined.

Remaining redundancies:

1.Introduction

Line/paragraph: 1–5

Issue: Repetitive explanation of population ageing and dementia rise. i.e.— first as a general factor in Latin America (line 2), then again to explain projections (line 3).

Suggested action: Combine and streamline for clarity. See this example: 'Cognitive impairment and dementia affect approximately 57 million people globally, with cases expected to exceed 150 million by 2050. In Latin America, projected population growth and ageing are anticipated to increase dementia prevalence by over 250% by 2050, compared to 75% in Western Europe.'

Line/paragraph: 11–13

Issue: Repetition of knowledge gap framing

Suggested action: Merge sentences

Line/paragraph: 14–18

Issue: Repetitive justification for simple tools in Colombia

Suggested action: Merge rationale

2. Results

Line/paragraph: 1–6

Issue: Redundant mention of participant numbers. Repeating “23,443 participants” twice within two sentences is redundant.

Suggested action: Combine sentence.See this example: 'The analysis included data from 23,443 of 23,694 (99%) SABE Colombia participants. Of these, 23,443 attempted the feet-together test, 20,943 the semi-tandem test, and 19,527 the tandem test, with complete covariate data.'

Line/paragraph: 7–10

Issue: Repeating prevalence rates already in abstract. Repeating prevalence figures for each test were already stated in the abstract and are repeated verbatim here without adding new context.

Suggested Action: Summarize or remove. See this example summarising without repeating: 'As shown in Table 1, prevalence of mild cognitive impairment decreased progressively across the three balance tests.'

if you wish to retain figures for clarity, state them once in the text or table, not both.

Reviewer #2: The Auther have addressed all the comment provided for him.

Regarding dual submission the author should confirm the manuscript is not under consideration elsewhere.

The availability of the data upon request must be clearly indicated

What are the requirements to access the data based on reasonable request from the publisher/Journal?

Expected response time. E.g. how many days it requires the author to get/access the data for this analysis

What are the potential barriers to access the data?

7. PLOS authors have the option to publish the peer review history of their article (what does this mean?). If published, this will include your full peer review and any attached files.

**Do you want your identity to be public for this peer review?** For information about this choice, including consent withdrawal, please see our Privacy Policy.

Reviewer #1: No

Reviewer #2: No

---

## [Editor Report · Decision Letter 2]

1 Sep 2025

Associations between 10-second standing balance tests and mild cognitive impairment in older adults in Colombia: a cross-sectional national survey

PGPH-D-25-00809R2

Dear Dr O'Donovan,

We are pleased to inform you that your manuscript 'Associations between 10-second standing balance tests and mild cognitive impairment in older adults in Colombia: a cross-sectional national survey' has been provisionally accepted for publication in PLOS Global Public Health.

Best regards,

Julia Robinson

Executive Editor